# Metabolomics Analysis of DRG and Serum in the CCI Model of Mice

**DOI:** 10.3390/brainsci13081224

**Published:** 2023-08-21

**Authors:** Kaimei Lu, Bin Fang, Yuqi Liu, Fangxia Xu, Chengcheng Zhou, Lijuan Wang, Lianhua Chen, Lina Huang

**Affiliations:** 1Department of Anesthesiology, Shanghai General Hospital, School of Medicine, Shanghai Jiao Tong University, Shanghai 200080, China; 18423483746@163.com (K.L.); 15905202716@163.com (Y.L.); xfx0103@sina.com (F.X.); 18895333049@163.com (C.Z.); novjul0815@163.com (L.W.); 2Department of Anesthesiology, Shanghai General Hospital, Nanjing Medical University, Shanghai 200080, China; bin.fang@shgh.cn

**Keywords:** neuropathic pain, metabolomics, DRG, serum

## Abstract

Neuropathic pain (NP) is a chronic and intractable disease that is widely present in the general population. It causes painful behavior and even mood changes such as anxiety and depression by altering the metabolism of substances. However, there have been limited metabolomics studies conducted in relation to neuropathic pain. Therefore, in this study, the effects of NP on metabolites in serum and the dorsal root ganglion (DRG) were investigated using a non-targeted metabolomics approach detected by gas chromatography–mass spectrometry (GC-MS) and liquid chromatography–mass spectrometry (LC-MS) to uncover differential metabolites and affected metabolic pathways associated with NP. Sixty mice were divided into the following two groups: a chronic constriction injury (CCI) of the sciatic nerve group and a sham group (*n* = 30, each). After 7 days of CCI modeling, the metabolite profiles of serum and the DRG were analyzed using GC/LC-MS for both the CCI and sham groups of mice. Multivariate analysis revealed differential metabolites and altered metabolic pathways between the CCI and sham groups. In the CCI group, our findings provided insights into the complex phospholipid, amino acid and acylcarnitine metabolic perturbations of DRG metabolism. In addition, phospholipid metabolic disorders and impaired glucose metabolism were observed in the serum. Moreover, the metabolic differences in the DRG and serum were correlated with each other. The results from this untargeted metabolomics study provide a perspective on the metabolic impact of NP on serum and the DRG.

## 1. Introduction

Neuropathic pain, which affects approximately 6.9–10% of the general population, is characterized by pain resulting from damage to or disease of the somatosensory nervous system [1,2]. It can arise from a variety of sources such as trauma, inflammation, tumors, metabolic disorders, neurodegenerative diseases, and viral infection. Depending on the site of injury, neuropathic pain can be divided into peripheral and central neuropathic pain [3]. Examples of peripheral neuropathic pain include pain following peripheral nerve injury, trigeminal neuralgia, and postherpetic neuralgia. Central neuropathic pain includes neuropathic pain after spinal cord or brain injury, multiple sclerosis, and chronic central post-stroke pain [4]. The manifestations of neuropathic pain include hyperalgesia, allodynia, and spontaneous pain, which significantly affects the overall well-being and quality of life of patients. Despite its prevalence, the underlying mechanisms of neuropathic pain remain poorly understood. Consequently, further exploration of the mechanisms of neuropathic pain is urgently needed to facilitate the development of more effective treatments. 

Metabolomics, a rapidly developing field within biological systems, has emerged as a new technique following advances in genomics and proteomics. It focuses primarily on the comprehensive analysis of low-molecular-weight endogenous metabolites, such as fatty acids, carbohydrates, amino acids, lipids, and vitamins in organisms [5]. This field has been extensively explored across a wide range of applications to capture metabolic changes associated with various disease states [6]. In addition, researchers can delve into the metabolic pathways of biological systems by looking at variations in metabolic components or contents. Currently, gas chromatography–mass spectrometry (GC-MS), liquid chromatography–mass spectrometry (LC-MS), and nuclear magnetic resonance spectroscopy (NMR) are the commonly employed technologies for non-targeted metabolomics [7]. Due to the complementary nature of LC-MS and GC-MS for metabolic analysis, the combination of LC-MS and GC-MS was used to carry out a more comprehensive analysis of the metabolic data in our study. 

The dorsal root ganglion (DRG) is an influential relay station for the processing and transmission of peripheral pain to the central nervous system, and it plays an essential role in the onset and maintenance of neuropathic pain [8]. In clinical work, neuropathic pain is mainly measured by patient’s subjective description and pain scale, but lacks objective indicators. Blood is the most commonly used test substance to react to the disease, which is less harmful to the human body. There have been numerous studies on DRG and serum organization in neuropathic pain, but few have explored metabolic changes in this condition and the connection between DRG and serum metabolites. Therefore, the purposes of this study are to explore the metabolite changes in the DRG and serum, then clarify whether there is a connection between the DRG and serum metabolite changes in neuropathic pain mice. The results will provide us with a new perspective for future clinical research. In this study, we used GC/LC-MS metabolomics techniques to detect the metabolites in the DRG and serum and used statistical methods to explore the differential metabolites and metabolic pathways in the CCI model. 

## 2. Materials and Methods

### 2.1. Animals

Sixty male C57BL/6 mice (8 weeks old) were purchased from the Laboratory Animal Center of Shanghai Jiao Tong University and divided into a CCI group and a sham group (*n* = 30, each group). All experimental protocols were approved by the Experimental Animal Management Ethics Committee of Shanghai General Hospital, Shanghai Jiao Tong University, School of Medicine (No. 2019AW009). In addition, all animal studies were performed in accordance with the ARRIVE guidelines. The mice were bred and maintained in a standard animal room with a 12 h day and 12 h night cycle, and given free access to food and water.

### 2.2. CCI Model

The chronic constriction injury of the sciatic nerve (CCI) model was established according to a previous study [9]. In brief, the mice were anesthetized by intraluminal injection of 10% chloral hydrate; then, the left sciatic nerve of the mice was carefully isolated under the mid-femur, and through the use of three segments of 5–0 chromic gut sutures to loosely ligate the sciatic nerve with 1 mm intervals. In the sham group, the sciatic nerve was exposed in the same way to the CCI mice but without ligations.

### 2.3. Behavior Test

The mechanical withdrawal threshold (MWT) of the mice was detected using von Frey filaments (NC12775, North Coast Medical Company, Carlsbad, CA, USA); the protocol was described previously [10]. Before testing, the mice were placed in an acrylic box on a metal mesh floor to be quiet for 30 min. Then, the von Frey filaments (0.16–2.0 g) were used to stimulate the plantar surface of the left hind paw vertically until the filament was bent for 6 s. Brisk withdrawal or paw flinching were considered to be positive responses. The MWT of mice was detected using the “up-down” method; the filament was started at 0.16 g and increased until the mice exhibited positive responses. Subsequently, lower-weight filaments were used to evaluate the sensory threshold.

The thermal withdrawal latency (TWL) of the mice was detected via the Hargreaves method (Ugo Basile SRL, Gemonio, VA, Italy). The mice were placed in a box on a glass platform to be quiet for 30 min. Heat was applied to stimulate the left hind paws of the mice; lifting or licking of the left hind paw was considered a pain response. The time from the onset of the heating stimulus to the appearance of the pain response was considered TWL. To avoid tissue damage, a 20 s cut-off time was set. Each mouse was tested 3 times with a 5 min interval between consecutive stimuli. The average value of 3 tests was taken as the TWL of the mice.

### 2.4. Sample Preparation

After 7 days of surgery, DRGs from the left L4–L6 segments of 10 mice from each group were collected together and used as one sample with a weight of approximately 20 mg. The blood of each mouse was collected separately by removing the eyeball after anesthesia and leaving it at room temperature for 30 min; then, the serum obtained was the supernatant of the blood after 1200 rpm centrifugation. The serum of six mice in the sham group and eight mice in the CCI group showed obvious hemolysis, therefore, the serum of these mice was eliminated. The serum of 5 mice (30 μL of each mouse) in each group was mixed to a 150 μL sample for detection. The protocol of the experiment is shown in Figure 1.

Approximately 20 mg of DRG tissue was accurately weighed and transferred to a 1.5 mL Eppendorf tube, and 600 μL mixed solution of methyl alcohol was added: water (4:1, *v*/*v*, containing 4 μg/mL L-2 chlorphenylalanine); then, the DRG sample was ground at 60 Hz for 2 min, sonicated in an ice-water bath for 10 min, and stored at −20 °C for 30 min; the extract was centrifuged at 12,000 rpm for 10 min. Approximately 150 μL of serum was added to a 1.5 mL Eppendorf tube with 2 μL of L-2-chlorophenylalanine (0.3 mg/mL) dissolved in methanol; then, the tube was vortexed to mix the contents; subsequently, 600 μL of an ice-cold mixture of methanol and acetonitrile (2/1, *v*/*v*) was added, and the mixture of serum was extracted via ultrasonic for 10 min in an ice-water bath; the extract was centrifuged at 12,000 rpm for 10 min. All steps were performed at 4 °C.

GC-MS: After centrifugation, the supernatant (150 μL) was dried in a freeze concentration centrifugal dryer. Approximately 80 μL of methoxamine hydrochloride pyridine solution (15 mg/mL) was added to the evaporated sample. The resultant mixture was vortexed for 2 min and incubated at 37 °C for 60 min. Then, the mixture with 50 μL of BSTFA and 20 μL of n-hexane was added. This mixture was subjected to vortexing for 2 min and then derivatized at 70 °C for 60 min. Subsequently, the samples were allowed to stand at ambient temperature for 30 min prior to their analysis using GC-MS. LC-MS: The supernatant (150 μL) from each sample was collected after centrifugation; then, the collected supernatant was filtered through 0.22 μm microfilters before being transferred to LC vials. These vials were then stored at −80 °C until the LC-MS analysis was performed.

### 2.5. Analysis Conditions of GC/LC-MS

GC-MS: An Agilent 7890B gas chromatography system (Agilent Technologies Inc., Santa Clara, California, USA) was used to analyze the samples. The separation of derivatives was achieved using a DB-5MS fused-silica capillary column from Agilent J & W Scientific (Folsom, CA, USA). Throughout the analysis, helium served as the carrier gas, with a constant flow rate of 1 mL/min through the column being maintained. To achieve optimal conditions for analysis, the injector temperature was maintained at 260 °C, and the injection volume was 1 μL. The chromatographic temperature program commenced with the initial oven temperature held at 60 °C for 0.5 min, followed by a gradual increase to 125 °C at a rate of 8 °C/min, subsequently to 210 °C at a rate of 5 °C/min, further to 270 °C at a rate of 10 °C/min, and finally reaching 305 °C at a rate of 20 °C/min, where it was held for 5 min. For the mass spectrometry (MS) analysis, the MS quadrupole temperature was set at 150 °C, and the ion source temperature was set at 230 °C. A collision energy of 70 eV was applied during the MS analysis. Mass spectrometric data were acquired in the full-scan mode, covering a range of *m*/*z* 50–500.

LC-MS: The metabolic profiling analysis was conducted using an ACQUITY UPLC I-Class system from Waters Corporation (Milford, CT, USA) in both electrospray ionization (ESI) positive and ESI negative ion modes. In the analysis, mobile phase A was composed of water and mobile phase B was a mixture of acetonitrile and methanol (2/3 *v*/*v*), with both containing 0.1% formic acid. Linear gradient: 1% B at 0 min, 30% B at 1 min, 60% B at 2.5 min, 90% B at 6.5 min, 100% B at 8.5 min, held at 100% B until 10.7 min, then returned to 1% B at 10.8 min, and finally maintained at 1% B until 13 min. During the analysis, the flow rate was set at 0.4 mL/min, and the column temperature was maintained at 45 °C. Data acquisition was carried out using a combination of the full scan mode, covering m/z ranges from 50 to 1000, and the MSE mode. The MSE mode involved acquiring two independent scans with different collision energies (CE) alternately during the analysis. The mass spectrometry parameters employed during the analysis were set as follows: A low-energy scan with a collision energy (CE) of 4 eV and a high-energy scan with a CE ramp ranging from 20 eV to 45 eV were applied to induce fragmentation of the ions. Argon gas was used as the collision-induced dissociation gas. For data acquisition, a scan time of 0.2 s was used, followed by an interscan delay of 0.02 s. The capillary voltage was set to 2.5 kV, and the cone voltage was maintained at 40 V. The source temperature was set to 115 °C, while the desolvation gas temperature was maintained at 450 °C. The desolvation gas flow rate was 900 L/h.

### 2.6. Data Preprocessing and Statistical Analysis of GC/LC-MS

The GC/MS data were processed using MS-DIAL software, which facilitated various analytical steps, including peak detection, peak identification, peak alignment, wave filtering, and missing value interpolation. By referencing the LUG database, metabolite characterization was achieved, and a data matrix was generated as a result. To ensure data reliability and accuracy, all peak signal intensities within each sample were segmented and normalized based on the standards, taking into account the relative standard deviation (RSD) greater than 0.3 after screening. The subsequent step of peak merging and redundancy removal was carried out, leading to the final formation of the data matrix.

The original LC-MS data underwent processing in Progenesis QI V2.3 software (Nonlinear Dynamics, Newcastle, UK). Subsequently, a qualitative analysis was performed by identifying the compounds using the Human Metabolome Database (HMDB), lipid maps, and self-built databases, based on the precise mass-to-charge ratio (*m*/*z*), secondary fragments, and isotopic distribution. Compounds with scores below 36 (out of 80) points were considered inaccurate and excluded. Subsequently, a data matrix was generated by combining the positive and negative ion data.

Data collected from LC-MS and GC-MS were used for the multivariate statistical analysis. To distinguish between the differing metabolites in the groups, orthogonal partial least-squares-discriminant analysis (OPLS-DA) was employed. The model’s validity was assessed using the model parameters R2X(cum), R2Y(cum), and Q2(cum). Additionally, SPSS 25.0 software was utilized for the univariate statistical analysis, including Student’s t-test and fold change (FC) analysis. To rank the overall contribution of each variable to group discrimination, variable importance of projection (VIP) values derived from the OPLS-DA model were utilized. Differential metabolites were selected based on VIP values greater than 1.0 and *p*-values less than 0.05, and their representation was displayed through a volcano map and a cluster map. Furthermore, the metabolic pathway enrichment analysis of these differential metabolites was conducted using the Kyoto Encyclopedia of Genes and Genomes (KEGG) pathway database.

## 3. Results

### 3.1. Behavioral Changes in the Mice

The baseline values of MWT and TWL on the day before surgery showed no significant difference between the sham and CCI groups. After 7 days of surgery, the mice in the CCI group exhibited heightened sensitivity to mechanical and thermal stimuli compared to those in the sham group (Figure 2). These results confirm the successful establishment of the CCI model.

### 3.2. Typical Metabolic Spectrums of the DRG and Serum

The total ion chromatograms of the DRG and serum samples were acquired by GC-MS, and the base peak chromatograms (BPIs) of the DRG and serum samples were acquired by LC-MS in the negative and positive ion modes (Appendix A). A strong signal response was observed in all samples, with a large number of metabolites detected over the entire run time.

### 3.3. The Differences between the CCI and Sham Groups in the DRG and Serum

To analyze the data distinction between the two groups, the OPLS-DA model was constructed to analyze the GC-MS and LC-MS data for the DRG and serum. In the OPLS-DA model score plots, there was significant separation both in the GC-MS and LC-MS data on the DRG and serum between the CCI and sham group (Figure 3). Both in the DRG and serum tissues, the model parameters R2X(cum), R2Y(cum), and Q2(cum) (Table 1) of the two models were all greater than 0.5 and close to 1, indicating that the fitting accuracy of the model was excellent and that the two groups had significant differences. Moreover, we found that the differences within each group were tiny, indicating that the metabolites of the mouse DRG and serum were different in the two groups.

### 3.4. The Differential Metabolites between the CCI and Sham Groups in the DRG and Serum

The different metabolites in the CCI and sham groups were analyzed by t-test, and volcano plots of the mice’s DRG and serum metabolites were produced to visually display the changed metabolites (Figure 4). There were 336 metabolites of the DRG and 350 metabolites of the serum in the two groups showing differences (*p* < 0.05). In addition, the VIP value obtained from the OPLS-DA was used to extract the significant differential metabolites. Notably, metabolites with VIP > 1.0, *p* < 0.05, and FC > 1 exhibited a significant increase, while metabolites with VIP > 1.0, *p* < 0.05, and FC < 1 showed a significant decrease. There were 51 significant differential metabolites identified in the mice DRG sample between the CCI and sham groups, including organic acids, lipids, and nucleosides (Table 2). Similarly, there were 44 significant differential metabolites also observed in the serum sample between the two groups, mainly including organic acids, lipids, carbohydrates and nucleosides (Table 3). In addition, the differential metabolites of the DRG and serum are visually displayed in heatmaps (Figure 5). The above results indicate that there were a large number of differential metabolites in the DRG and serum in the CCI group and sham group. 

### 3.5. The Differential Metabolic Pathways between the CCI and Sham Groups in the DRG and Serum

Metabolic pathway enrichment analysis of differential metabolites was performed based on the KEGG database. Data on differential metabolites were imported into the KEGG database to investigate the weights of the metabolic pathways. The enrichment map and bubble chart of the metabolic pathways showed the top 20 differential metabolic pathways of the DRG (Figure 6A,B) and serum (Figure 6C,D). In the DRG, there were seven significantly different metabolic pathways between the sham and CCI groups (raw *p* < 0.05, impact > 0.05), mainly including the alanine, aspartate, and glutamate metabolisms; arginine biosynthesis; central carbon metabolism; and autophagy and choline metabolism, and details on the metabolic pathway analysis are presented in Table 4. In the serum, there were 18 significantly different metabolic pathways between the sham and CCI groups (raw *p* < 0.05; impact > 0.05), mainly including the pentose phosphate pathway, choline metabolism, glycerophospholipid metabolism, central carbon metabolism, and so forth, and details on the metabolic pathway analysis are presented in Table 5.

## 4. Discussion

In this study, we explored the abnormal metabolism of the DRG and serum in CCI models using an untargeted metabolomics approach. We found 51 metabolites in the DRG and 44 metabolites in serum that appeared to be significantly altered in the CCI group. These differential metabolites mainly include organic acids, carbohydrates, lipids and lipid-like molecules, nucleosides, and others. 

In the DRG metabolites, we reported reductions in metabolites associated with phospholipid biosynthesis. Phospholipids are the main constituents of the phospholipid bilayer of the cell membrane and maintain the normal activity and function of the cell. At the same time, phospholipids are also vital in mitochondrial function and myelin maintenance [11,12,13]. PC is a phospholipid with elevated content in peripheral myelin [14]. The myelin sheath is an important component of neurons, which plays a dual role in promoting nerve impulse conduction and preventing axonal injury [15]. Phospholipids contains phosphatidyl choline (PC), phosphatidyl ethanolamine (PE), phosphatidyl serine (PS), phosphatidyl inositol (PI), phosphatidyl glycerol (PG), and so on. In this study, we found that some PC, PE and PS contents were significantly downregulated in the CCI models. Additionally, PE-NME, a precursor of phosphatidic choline synthesis, and phosphatidic acid (PA), a precursor to phospholipid synthesis, were also significantly downregulated (Figure 7A). However, the opposite result was observed in the serum, where the expression of sectional PC, PE, and PI was significantly upregulated, and the precursor substances for phospholipid synthesis, PA and PE-NME, were significantly upregulated (Figure 7A). Moreover, among the differential metabolites, there were four identical metabolites that showed opposite trends in DRG and serum (Figure 7B), which may be related to the metabolites in the DRG entering the blood, suggesting that some blood metabolites can reflect metabolic changes in the DRG. The above results indicate that NP may affect mitochondrial function or myelin maintenance by inhibiting phospholipid synthesis in the DRG, and studies have shown that structural abnormalities in either the mitochondria or myelin sheaths can cause pain behavior [16,17]. Therefore, NP may cause neuronal dysfunction and hyperalgesia by inhibiting phospholipid synthesis in the DRG. The opposite expression trends of differential metabolites in serum and the DRG suggest that there exists a close correlation between metabolic disorders of the blood and the DRG.

Among the differential metabolites identified in the DRG, four amino acids, N-Acetyl-L-Aspartic Acid (NAA), N-Acetyl-L-Glutamic Acid, N-Acetyl-1-aspartylglutamic acid (NAAG), and L-Aspartic acid, were significantly decreased (Figure 7C). NAA, a marker of neuronal density and viability, is mainly synthesized in neurons [18]. A decreased NAA concentration suggests the possible neuronal damage or dysfunction, and several studies have identified decreased NAA as a clinical indicator of neurological dysfunction [19,20]. Additionally, some studies have shown that the NAA reduction was also observed in the brain of patients with neuropathic pain [19,21], and the NAA level was positively correlated with the severity of neuropathic pain symptoms. In this study, we observed a decreased NAA expression in the DRG of CCI mice via metabolomics, suggesting that DRG neurons in NP mice may be damaged or dysfunctional. Studies have implied that NAA is synthesized from ASP in mitochondria, and then transported to the Schwann cells to synthesize myelin [22]. However, the decrease in NAA in the DRG suggests a possible mechanism for demyelination in NP. NAAG is the third most common neurotransmitter in the nervous system behind glutamate and GABA [23]. It inhibits the release of glutamate through the activation of metabotropic glutamate mGlu3 receptors at the presynaptic membrane and stimulates the release of neuroprotective growth factors through the activation of mGlu3 receptors on glial cells [24]. Therefore, NAAG exerts neuroprotective effects by inhibiting excitotoxicity. Furthermore, it has been demonstrated that the inhibition of NAAG degradation can be used for the treatment of several diseases, such as animal models of traumatic brain injury and neuropathic pain [24]. The above indicates that NAGG has a protective effect on neurons. In this study, we found a significant decrease in NAAG in the DRG of the CCI mice, suggesting that it may reduce the inhibition of excitotoxicity to promote the progression of NP. Similarly, KEGG enrichment analysis showed significant changes in the alanine, aspartate, and glutamate metabolic pathways involved by NAA, NAAG, and L-aspartate (Table 4). Studies have shown that the depletion of amino acids in the tissues suggests cell death or neurodegeneration [25,26,27]. We observed a significant decrease in amino acids in the DRG of CCI model, which suggested that cell death or neural degeneration may occur in DRGs, but further studies are needed to confirm this.

In the DRG metabolites, we observed a significant increase in oleic acid (Figure 7C). Oleic acid, as a known endogenous neurotrophic factor, combined with albumin and suggests neuroprotective effects by inhibiting nociceptive reflexes in SCI, resulting in neuroprotective effects [28]. One study has shown that extracellular albumin and oleic acid, when entering neurons, can promote dendritic growth [29,30,31] and also promote sphingomyelin synthesis [32,33]. Furthermore, oleic acid inhibits the production of proinflammatory mediators in microglia cultures [34] and can attenuate the pain response in patients with arthritis [35]. Therefore, oleic acid plays an important role in repairing nerve damage and attenuating pain responses. In this study, we found a significant increase in oleic acid in the CCI model, which could be a self-healing process after nerve injury, but its role in the DRG needs further investigation. 

Furthermore, we observed a significant reduction in acylcarnitine levels in the DRG of the CCI group (Figure 7D). Acylcarnitine is located in the inner mitochondrial membrane, and its role is to transport the acyl of fatty acids from the cytoplasm to the mitochondria, so as to perform β-oxidation to produce energy; it is an important part of the energy metabolism [36,37]. In our study, we found a significant decrease in acylcarnitine levels in the CCI group, suggesting that NP may cause an energy metabolism disorder of the fatty acids in the DRG that affects the supply of energy to the cell. 

In the serum metabolites, we found that there were significant differences in most of the carbohydrates, such as glucose, between the sham group and the CCI group (Figure 7E), and metabolic pathway analysis showed that the pentose phosphate pathway was the most significant differential metabolic pathway between the sham and CCI groups (Figure 7C). In addition to providing energy, the pentose phosphate pathway also provides raw materials for the synthesis of various substances. It has been shown that the brain glucose metabolism is increased in rats with chronic pain [38,39], and a clinical investigation showed a significant association between patients with chronic pain and diabetes [40]. In our study, we observed a significant increase in serum glucose in the CCI mice, suggesting that the increase in glucose may be caused by increased energy metabolism or the eating changes in the mice with pain behavior. In addition, the increased serum glucose further aggravated the pain response in turn. It is known that patients with diabetes suffer from peripheral neuropathic pain, which is caused by peripheral neuropathy caused by increased blood glucose [41]. Studies have shown that diabetes leads to peripheral neuropathy by reducing phospholipid synthesis to induce myelin abnormalities in nerves [42,43]. In this study, serum glucose and other carbohydrates of serum in the CCI group showed significant increases in the CCI group, suggesting that NP may lead to abnormal energy metabolism and elevated blood glucose in the serum; elevated blood glucose may further inhibit phospholipid synthesis and finally exacerbate the pain response.

There are some limitations in our study: only male mice were used in this experiment to study the metabolite changes in DRG and serum in neuropathic pain, and the effect of gender on pain and metabolites was not explored. Further experiments are needed to test this. In this study, we simply observed the results of metabolite changes in the DRG and serum of NP mice, but failed to further explore the reasons for metabolite changes caused by NP, which is a limitation of this experiment. It has been shown that dietary strategies influence pain behavior in patients with chronic pain [44,45]. And lifestyle interventions may alter the gut microbiome and affect the pain response [46]. Therefore, we guess that in addition to the direct nerve damage, the stress associated with the nerve injury and the changes in the animal’s activity, energy metabolism, and diet in turn affect the metabolites. These points require further study. 

## 5. Conclusions

Through untargeted metabolomics tested by GC-MS and LC-MS, our study found that peripheral nerve injury caused dramatic changes in the DRG and serum metabolites and the major metabolic pathway changes. Taken together, peripheral nerve injury results in significant alterations in phospholipid, amino acid, oleic acid, and acylcarnitine in the DRG and alterations in glucose in serum. Furthermore, some metabolites showed opposite trends in the DRG and serum, which implicated a close relationship between DRG and serum metabolites, and serum metabolites could reflect the metabolic changes in the DRG. In addition, the reasons for metabolite changes could be as follows: direct nerve damage, the stress associated with the nerve injury and the changes in the animals’ activity, and energy metabolism and diet affected by pain behavior. Moreover, the changed metabolites can reflect cellular structural disruption or abnormal energy metabolism, as well as postinjury self-protection. There are some limitations to this experiment, and structural and functional changes in tissues still need to be validated by further histomorphology and molecular biology studies.

## Figures and Tables

**Figure 1 brainsci-13-01224-f001:**
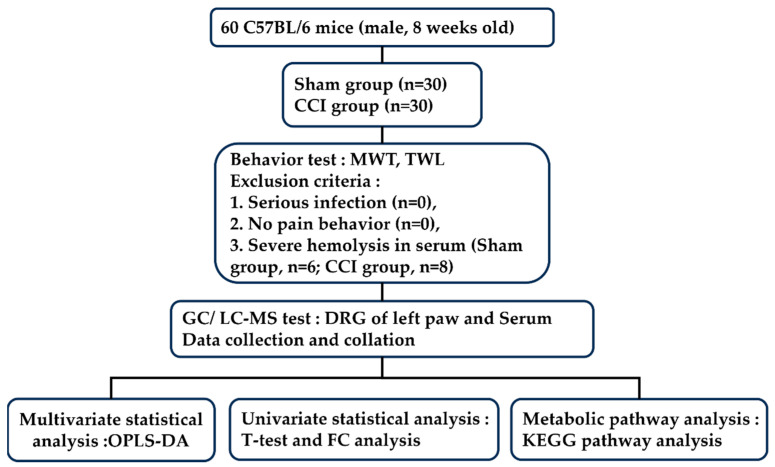
The flow chart of experiment.

**Figure 2 brainsci-13-01224-f002:**
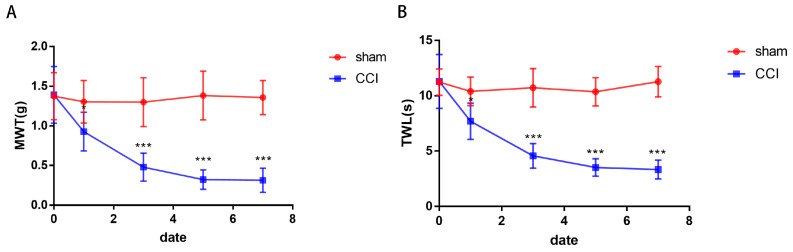
The behavioral changes in the mice. Compared to the sham group, the MWT (**A**) and TWL (**B**) of the CCI mice significantly decreased after surgery (*n* = 6, * *p* < 0.05, *** *p* < 0.001).

**Figure 3 brainsci-13-01224-f003:**
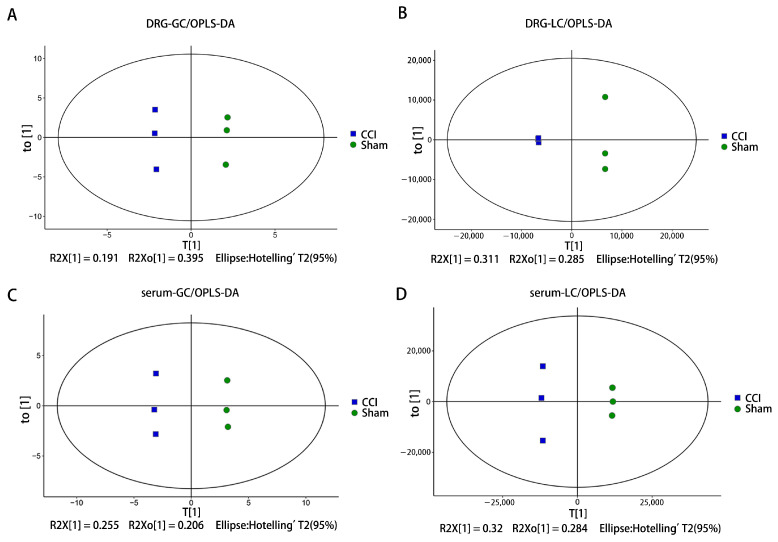
OPLS-DA score plot of LC and GC in the DRG and serum. (**A**) OPLS-DA score plot of the DRG-GC. (**B**) OPLS-DA score plot of the DRG-LC. (**C**) OPLS-DA score plot of the serum-GC. (**D**) OPLS-DA score plot of the serum-LC. DRG-GC and serum-GC: the GC-MS data of the DRG or serum. DRG-LC, serum-LC, the LC-MS data of the DRG or serum.

**Figure 4 brainsci-13-01224-f004:**
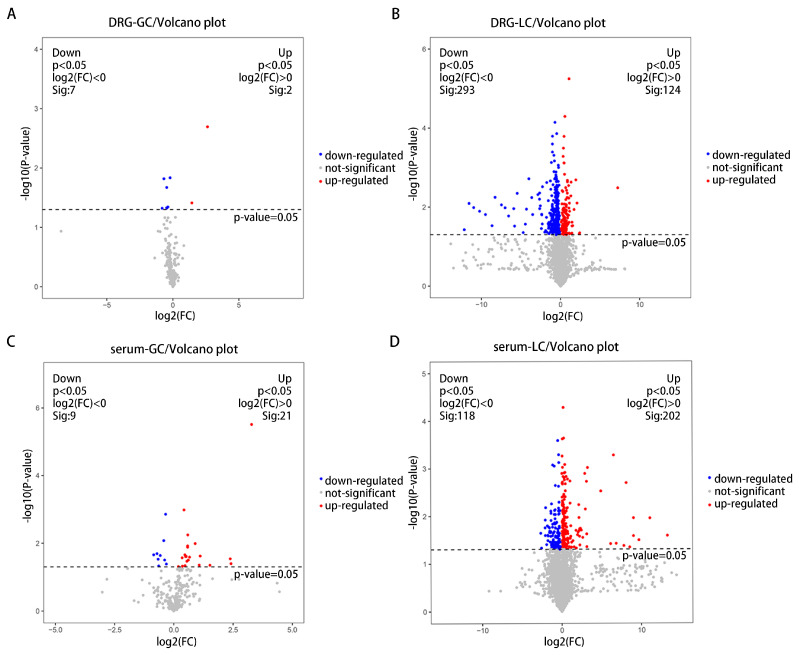
Metabolites of the DRG and serum detected by GC-MS and LC-MS. CCI vs. the sham. (**A**) Differential metabolites between two groups, based on GC-MS data from DRG. (**B**) Differential metabolites between two groups, based on LC-MS data from DRG. (**C**) Differential metabolites between two groups, based on GC-MS data from serum. (**D**) Differential metabolites between two groups, based on LC-MS data from serum. Blue dots: compared to sham group, down-regulated metabolites in CCI group, *p* < 0.05 and fold change < 1. Red dots: compared to sham group, up-regulated metabolites in CCI group, *p* < 0.05 and fold change > 1.

**Figure 5 brainsci-13-01224-f005:**
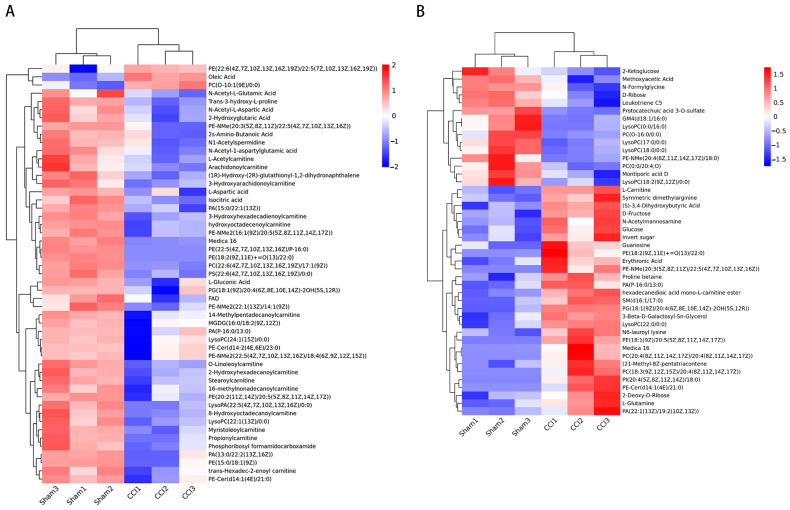
Heatmap of differential metabolites between the CCI and sham groups in the DRG and serum. (**A**) Heatmap of differential metabolites in the DRG. (**B**) Heatmap of differential metabolites in serum. The colors indicate the expression abundance of the metabolites.

**Figure 6 brainsci-13-01224-f006:**
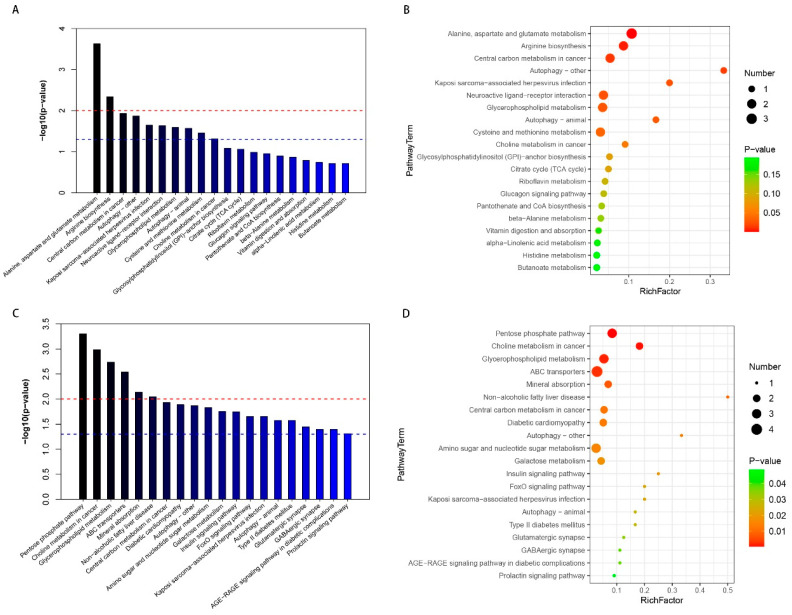
Differential metabolic pathways analysis between the CCI and sham groups. The differential metabolites were subjected to metabolic pathway enrichment analysis, with *p* < 0.05, VIP > 1; top 20 related metabolic pathways were analyzed. (**A**,**C**) Enrichment map: the red dummy line indicates a *p*-value of 0.01, and the blue dummy line indicates a *p*-value of 0.05. Signal pathways higher than the blue dummy line represent significant differences. (**B**,**D**) Bubble chart: the size of the dots represents the number of metabolites. RichFactor, the number of differential metabolites/total number of metabolites in this pathway.

**Figure 7 brainsci-13-01224-f007:**
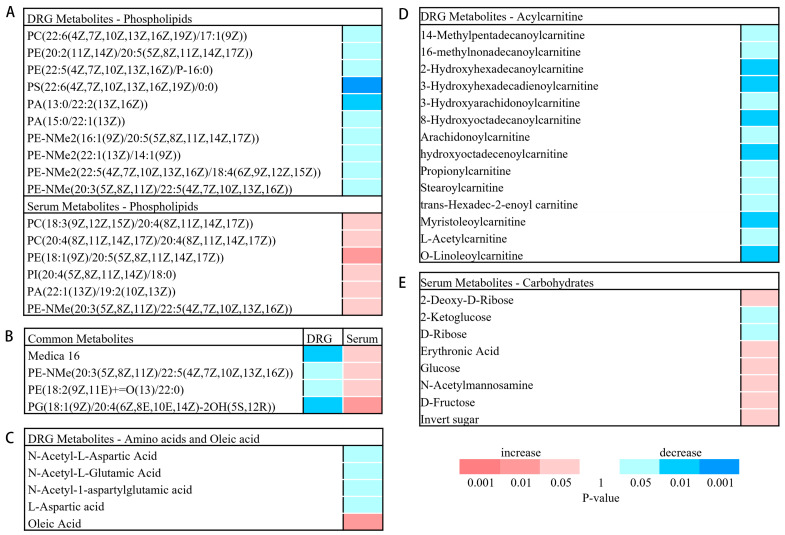
Differential expression of metabolites in the DRG and serum. (**A**,**B**) Differential expression of phospholipids in the DRG and serum. (**C**) Amino acids and oleic acid expressions in the DRG. (**D**) Acylcarnitine expression in the DRG. (**E**) Carbohydrates expression of serum. The colors represent the *p*-value, with red representing upregulation and blue representing a decrease. D, DRG. S, serum.

**Table 1 brainsci-13-01224-t001:** Model parameters for the DRG.

	DRG		Serum	
Parameters	GC-MS	LC-MS	GC-MS	LC-MS
R2X(cum)	0.720	0.730	0.718	0.781
R2Y(cum)	1.000	1.000	1.000	1.000
Q2(cum)	0.887	0.932	0.848	0.879

R2X(cum), explanation rate in the X-axis. R2Y(cum), explanation rate in the Y-axis. Q2(cum), cumulative prediction rate.

**Table 2 brainsci-13-01224-t002:** List of DRG metabolites with significant changes in the CCI and sham groups.

	Metabolites	VIP	*p*-Value	log2 (FC)	FC	Data Class
Organic acids and derivatives	N-Acetyl-L-Aspartic Acid (NAA)	2.032	0.015	−0.679	0.625	GC
	2s-Amino-Butanoic Acid	1.643	0.047	−0.478	0.718	GC
	N-Acetyl-L-Glutamic Acid	1.452	0.045	−0.373	0.772	GC
	L-Gluconic Acid	1.586	0.047	−0.459	0.727	GC
	2-Hydroxyglutaric Acid	1.662	0.021	−0.467	0.724	GC
	N-Acetyl-1-aspartylglutamic acid (NAAG)	7.209	0.022	−0.502	0.706	LC
	L-Aspartic acid	4.407	0.046	−0.447	0.734	LC
	(1R)-Hydroxy-(2R)-glutathionyl-1,2-dihydronaphthalene	2.331	0.007	−0.524	0.695	LC
	Trans-3-hydroxy-L-proline	1.152	0.002	−0.529	0.693	LC
	N1-Acetylspermidine	1.236	0.040	−0.518	0.698	LC
	Isocitric acid	4.202	0.006	−0.412	0.752	LC
Lipids and lipid-like molecules	Oleic Acid	4.172	0.002	2.623	6.160	GC
	14-Methylpentadecanoylcarnitine	8.562	0.044	−0.346	0.787	LC
	16-methylnonadecanoylcarnitine	2.403	0.027	−0.547	0.685	LC
	2-Hydroxyhexadecanoylcarnitine	9.042	0.008	−0.361	0.778	LC
	3-Hydroxyhexadecadienoylcarnitine	1.165	0.001	−0.835	0.561	LC
	3-Hydroxyarachidonoylcarnitine	1.430	0.016	−0.488	0.713	LC
	8-Hydroxyoctadecanoylcarnitine	6.045	0.007	−0.426	0.744	LC
	Arachidonoylcarnitine	5.844	0.043	−0.354	0.782	LC
	hydroxyoctadecenoylcarnitine	2.967	0.003	−0.473	0.721	LC
	Propionylcarnitine	3.042	0.013	−0.839	0.559	LC
	Stearoylcarnitine	9.281	0.010	−0.416	0.749	LC
	trans-Hexadec-2-enoyl carnitine	4.306	0.027	−0.436	0.739	LC
	MGDG(16:0/18:2(9Z,12Z))	5.015	0.009	−2.552	0.171	LC
	Myristoleoylcarnitine	3.411	0.007	−0.546	0.685	LC
	L-Acetylcarnitine	10.425	0.038	−0.643	0.640	LC
	LysoPA(22:5(4Z,7Z,10Z,13Z,16Z)/0:0)	1.847	0.015	−3.526	0.087	LC
	LysoPC(22:1(13Z)/0:0)	1.746	0.012	−1.101	0.466	LC
	LysoPC(24:1(15Z)/0:0)	2.291	0.027	−4.447	0.046	LC
	O-Linoleoylcarnitine	6.888	0.002	−0.633	0.645	LC
	PA(13:0/22:2(13Z,16Z))	1.508	0.030	−8.663	0.002	LC
	PA(15:0/22:1(13Z))	1.221	0.003	−0.955	0.516	LC
	PA(P-16:0/13:0)	7.741	0.027	−2.308	0.202	LC
	PC(22:6(4Z,7Z,10Z,13Z,16Z,19Z)/17:1(9Z))	6.316	0.015	−9.492	0.001	LC
	PE(15:0/18:1(9Z))	2.907	0.010	−10.981	0.000	LC
	PE(20:2(11Z,14Z)/20:5(5Z,8Z,11Z,14Z,17Z))	2.908	0.037	−12.171	0.000	LC
	PE(22:5(4Z,7Z,10Z,13Z,16Z)/P-16:0)	1.239	0.046	−23.919	0.000	LC
	PE(22:6(4Z,7Z,10Z,13Z,16Z,19Z)/22:5(7Z,10Z,13Z,16Z,19Z))	2.854	0.024	1.446	2.725	LC
	PE-Cer(d14:1(4E)/21:0)	4.508	0.044	−4.718	0.038	LC
	PE-Cer(d14:2(4E,6E)/23:0)	1.866	0.011	−2.636	0.161	LC
	PE-NMe(20:3(5Z,8Z,11Z)/22:5(4Z,7Z,10Z,13Z,16Z))	1.501	0.013	−10.238	0.001	LC
	PE-NMe2(16:1(9Z)/20:5(5Z,8Z,11Z,14Z,17Z))	1.821	0.030	−5.815	0.018	LC
	PE-NMe2(22:5(4Z,7Z,10Z,13Z,16Z)/18:4(6Z,9Z,12Z,15Z))	2.677	0.047	−1.459	0.364	LC
	PE-NMe2(22:1(13Z)/14:1(9Z))	1.060	0.018	−0.304	0.810	LC
	PC(O-10:1(9E)/0:0)	1.308	0.009	0.727	1.656	LC
	PS(22:6(4Z,7Z,10Z,13Z,16Z,19Z)/0:0)	2.492	0.001	−0.257	0.837	LC
	Medica 16	6.398	0.009	−7.468	0.006	LC
	Phosphoribosyl formamidocarboxamide	2.393	0.026	−1.570	0.337	LC
Nucleosides	FAD	1.949	0.043	−0.217	0.860	LC
and	PE(18:2(9Z,11E)+=O(13)/22:0)	1.546	0.031	−24.503	0.000	LC
others	PG(18:1(9Z)/20:4(6Z,8E,10E,14Z)-2OH(5S,12R))	1.768	0.005	−2.866	0.137	LC

VIP, variable important in projection; *p*-value, significance of difference; FC, fold change.

**Table 3 brainsci-13-01224-t003:** List of serum metabolites with significant changes in the CCI and sham groups.

	Metabolites	VIP	*p*-Value	log2 (FC)	FC	Data Class
Organic acids and derivatives	(S)-3,4-Dihydroxybutyric Acid	1.433	0.006	0.589	1.505	GC
	L-Glutamine	3.172	0.029	2.380	5.204	GC
	Methoxyacetic Acid	1.654	0.022	−0.850	0.555	GC
	N-Formylglycine	1.362	0.023	−0.573	0.672	GC
	L-Carnitine	6.030	0.002	0.481	1.396	LC
	Leukotriene C5	4.168	0.019	−0.484	0.715	LC
	Proline betaine	1.449	0.014	0.517	1.431	LC
	Protocatechuic acid 3-O-sulfate	1.048	0.001	−0.911	0.532	LC
	Symmetric dimethylarginine	1.136	0.017	0.617	1.533	LC
Carbohydrates	2-Deoxy-D-Ribose	2.222	0.045	1.521	2.870	GC
	2-Ketoglucose	1.385	0.047	−0.638	0.642	GC
	D-Ribose	1.101	0.031	−0.389	0.764	GC
	Erythronic Acid	1.405	0.032	0.611	1.527	GC
	Glucose	1.478	0.026	0.663	1.583	GC
	N-Acetylmannosamine	1.274	0.025	0.496	1.410	GC
	D-Fructose	5.918	0.010	0.339	1.265	LC
	Invert sugar	2.296	0.037	0.410	1.328	LC
Lipids and lipid-like	3-Beta-D-Galactosyl-Sn-Glycerol	1.400	0.012	0.582	1.497	GC
	(21-Methyl-8Z-pentatriacontene	1.526	0.022	0.426	1.344	LC
molecules	GM4(d18:1/16:0)	11.617	0.023	−0.527	0.694	LC
	hexadecanedioic acid mono-L-carnitine ester	2.327	0.001	2.928	7.612	LC
	LysoPC(0:0/16:0)	20.970	0.024	−0.415	0.750	LC
	LysoPC(17:0/0:0)	1.008	0.009	−0.377	0.770	LC
	LysoPC(18:0/0:0)	5.459	0.016	−0.301	0.812	LC
	LysoPC(18:2(9Z,12Z)/0:0)	2.495	0.037	−0.377	0.770	LC
	LysoPC(22:0/0:0)	2.696	0.006	2.161	4.473	LC
	Medica 16	1.381	0.042	7.823	226.438	LC
	Montiporic acid D	1.224	0.036	−0.544	0.686	LC
	N6-lauroyl lysine	1.290	0.007	0.278	1.213	LC
	PA(22:1(13Z)/19:2(10Z,13Z))	1.278	0.048	1.090	2.129	LC
	PA(P-16:0/13:0)	2.816	0.020	2.471	5.543	LC
	PC(0:0/20:4;O)	1.087	0.027	−0.581	0.668	LC
	PC(18:3(9Z,12Z,15Z)/20:4(8Z,11Z,14Z,17Z))	4.044	0.026	13.300	###	LC
	PC(20:4(8Z,11Z,14Z,17Z)/20:4(8Z,11Z,14Z,17Z))	3.861	0.032	9.718	842.057	LC
	PC(O-16:0/0:0)	1.592	0.034	−0.329	0.796	LC
	PE(18:1(9Z)/20:5(5Z,8Z,11Z,14Z,17Z))	5.844	0.002	8.141	282.319	LC
	PE-Cer(d14:1(4E)/21:0)	1.536	0.039	26.618	###	LC
	PE-NMe(20:3(5Z,8Z,11Z)/22:5(4Z,7Z,10Z,13Z,16Z))	1.784	0.011	11.105	2202.369	LC
	PE-NMe(20:4(8Z,11Z,14Z,17Z)/18:0)	5.221	0.018	−1.535	0.345	LC
	PI(20:4(5Z,8Z,11Z,14Z)/18:0)	5.069	0.011	9.040	526.430	LC
	SM(d16:1/17:0)	4.910	0.007	2.992	7.956	LC
Nucleosides and others	Guanosine	3.322	0.040	2.412	5.324	GC
	PE(18:2(9Z,11E)+ = O(13)/22:0)	1.728	0.038	6.857	115.920	LC
	PG(18:1(9Z)/20:4(6Z,8E,10E,14Z)-2OH(5S,12R))	1.628	0.001	3.281	9.717	LC

VIP, variable important in projection; *p*-value, significance of difference; FC, fold change; ### >10,000.

**Table 4 brainsci-13-01224-t004:** The details of the metabolic pathways of the DRG.

Pathway Name	Hit	Match Status	*p*-Value	FDR	Rich Factor
Alanine, aspartate, and glutamate metabolism	N-Acetyl-L-Aspartic Acid, N-Acetyl-1-aspartylglutamic acid, L-Aspartic acid	3/38	0.00023	0.00796	0.10714
Arginine biosynthesis	N-Acetyl-L-Glutamic Acid, L-Aspartic acid	2/23	0.00462	0.07847	0.08696
Central carbon metabolism in cancer	L-Aspartic acid, Isocitric acid	2/37	0.01171	0.11442	0.05405
Autophagy—other	PE (polytypes)	1/3	0.01355	0.11442	0.33333
Kaposi sarcoma-associated herpesvirus infection	PE (polytypes)	1/5	0.02248	0.11442	0.20000
Neuroactive ligand–receptor interaction	N-Acetyl-1-aspartylglutamic acid, L-Aspartic acid	2/53	0.02323	0.11442	0.03774
Glycerophospholipid metabolism	PE, LysoPC (polytypes)	2/56	0.02575	0.11442	0.03571
Autophagy—animal	PE (polytypes)	1/6	0.02692	0.11442	0.16667
Cysteine and methionine metabolism	2s-Amino-Butanoic Acid, L-Aspartic acid	2/66	0.03494	0.13200	0.03030
Choline metabolism in cancer	LysoPC (polytypes)	1/11	0.04884	0.16605	0.09091

Row *p*, the *p* value calculated from the enrichment analysis; FDR, false discovery rate.

**Table 5 brainsci-13-01224-t005:** The details of the metabolic pathways of serum.

Annotation	Hit	Match Status	Row *p*	FDR	Rich Factor
Pentose phosphate pathway	2-Deoxy-D-Ribose, Glucose, D-Ribose	3/36	0.00050	0.02744	0.08333
Choline metabolism in cancer	LysoPC, PC (polytypes)	2/11	0.00104	0.02849	0.18182
Glycerophospholipid metabolism	LysoPC, PC, PE (polytypes)	3/56	0.00183	0.03360	0.05357
Mineral absorption	Glucose, L-Glutamine	2/29	0.00729	0.08020	0.06897
Non-alcoholic fatty liver disease	Glucose	1/2	0.00905	0.08171	0.50000
Central carbon metabolism in cancer	Glucose, L-Glutamine	2/37	0.01171	0.08171	0.05405
Diabetic cardiomyopathy	Glucose, L-Carnitine	2/39	0.01296	0.08171	0.05128
Autophagy—other	PE (polytypes)	1/3	0.01355	0.08171	0.33333
Insulin signaling pathway	Glucose	1/4	0.01803	0.08262	0.25000
FoxO signaling pathway	Glucose	1/5	0.02248	0.08833	0.20000
Kaposi sarcoma-associated herpesvirus infection	Glucose	1/5	0.02248	0.08833	0.20000
Autophagy—animal	Glucose	1/6	0.02692	0.09255	0.16667
Type II diabetes mellitus	Glucose	1/6	0.02692	0.09255	0.16667
Glutamatergic synapse	Glucose	1/8	0.03575	0.11565	0.12500
GABAergic synapse	Glucose	1/9	0.04013	0.11616	0.11111
AGE-RAGE signaling pathway in diabetic complications	Glucose	1/9	0.04013	0.11616	0.11111
Prolactin signaling pathway	Glucose	1/11	0.04884	0.13431	0.09091

Row *p*, the *p* value calculated from the enrichment analysis; FDR, false discovery rate.

## Data Availability

The data and analyses used in this study can be obtained from the corresponding author with a reasonable request.

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
