# Peer review of "Metabolomics Analysis of DRG and Serum in the CCI Model of Mice"

_brainsci, 2023, doi:10.3390/brainsci13081224_

Round 1

Reviewer 1 Report

Comments and Suggestions for Authors

The study titled “Metabolomics analysis of DRG and serum in the CCI model of mice” by Lu et al. aimed to unravel the metabolomic differences between neuropathic pain (NP) induced by CCI and sham animals. To achieve this, they used two techniques: GC-MS and LC-MS. The authors used two behavioral tests to demonstrate that mice under CCI experienced NP in comparison to sham mice. By analyzing DRG extracts and blood serum, the study revealed that NP caused significant changes in DRG and serum metabolites, affecting several metabolic pathways. The results are presented clearly, and the statistical analyses appear appropriate for this investigation. The findings are interesting and provide valuable data for explaining how sciatic nerve injury leads to hyperalgesia. The study highlights disorders in phospholipid, amino acid, and energy metabolism in DRG, most of which counteract the observed effects in the serum. This suggests detrimental malfunctioning of myelinization of peripheral nerves. The results are very interesting and detailed with a clear effect comparing the NP mice group with the sham group, helping the comprehension of peripheral pain in this model.

The results are very interesting and detailed with a clear effect comparing the NP mice group with the sham group, helping the comprehension of peripheral pain on this model.

However, a few amendments, fixes, or clarifications are still needed:

1.       The size of parameters, axis labels, and labels in general of Figure 2 are too small. Please increase it so that they can be readable with no additional help.

2.       Line 85: Behavioral test

3.       Figure 5 is outside margins and cannot be read it properly

4.       Please, fix the margins of Tables 4 and 5 so that they fit

5.       Line 267: What means by anomalous metabolism??

6.       The subtitles in Figure 6 are confusing: "DRG metabolites" are constantly repeated. Please rename it accordingly with the figure legend.

7.       Line 303: What could be the reason for the significant enrichment of metabolites related to Alanine, aspartate, and glutamate metabolism? Please explain

8.       Line 320: There should say “oleic acid…it suggests…”

9.       The authors need to explain the limitations of this study more in detail. Why are you using only males? Would you expect to find differences between males and females? would you consider exploring differences between male/men or female/women?

10.   Is the increased glucose measured in CCI animals caused by the sciatic nerve procedure? I understand that, in diabetic patients, the excess of glucose alters fatty acid biosynthesis, as has been shown, but the CCI procedure is not causing diabetes per se. Then, the question is what causes the increment in glucose. Can you extend your argument in your discussion?

Comments on the Quality of English Language

The quality of English is good. 

Author Response

Dear reviewer,

Thank you very much for your valuable comments. We are very pleased with your interest in our research, according to your comments, we have revised the manuscript and explained your questions. All the revisions are highlighted in red, please check them.

  1. The size of parameters, axis labels, and labels in general of Figure 2 are too small. Please increase it so that they can be readable with no additional help.

Response:

Thanks a lot for your comments, we have increased and bolded the size of parameters and labels in general of Figure 3 (original Figure 2) on page 6, and marked as red.

  1. Line 85: Behavioral test

Response:

We are so sorry for our carelessness, we have revised the word “Behavior test” as “Behavioral test” on line 90, and marked as red.

  1. Figure 5 is outside margins and cannot be read it properly

Response:

Thanks for your helpful comments, and we resized Figure 6(original Figure 5) to make it readable on page 11, and marked as red.

  1. Please, fix the margins of Tables 4 and 5 so that they fit

Response:

Thanks for your comments, and we revised the margins of Tables 4 and 5 to make it fit on page 7-10, and marked as red.

  1. Line 267: What means by anomalous metabolism??

Response:

We are so sorry for our mistake of writing “abnormal metabolism” as “anomalous metabolism”, and we have revised the word to “abnormal metabolism” on line 288, and marked as red.

  1. The subtitles in Figure 6 are confusing: "DRG metabolites" are constantly repeated. Please rename it accordingly with the figure legend.

Response:

Thanks a lot for your comments, we have revised the subtitles in Figure 7(original Figure 6) on page 13, and marked as red.

  1. Line 303: What could be the reason for the significant enrichment of metabolites related to Alanine, aspartate, and glutamate metabolism? Please explain

Response:

Thank you so much for your valuable comments. And we have provided a further discussion for the significant enrichment of metabolites related to Alanine, aspartate, and glutamate metabolism on line 322-346, and marked as red. The content is as follows:

“Among the differential metabolites identified in DRG, there were 4 amino acid, N-Acetyl-L-Aspartic Acid (NAA), N-Acetyl-L-Glutamic Acid, N-Acetyl-1-aspartylglutamic acid (NAAG) and L-Aspartic acid, significantly decreased (Figure 7C). NAA, a marker of neuronal density and viability, is mainly synthesized in neurons [19]. A decreased NAA concentration suggests the possible neuronal damage or dysfunction, and several studies have identified decreased NAA as a clinical indicator of neurological dysfunction [20, 21]. Additionally, some studies have shown that the NAA reduction was also observed in the brain of patients with neuropathic pain [20, 22], and the NAA level was positively correlated with the severity of neuropathic pain symptoms. In this study, we observed the decreased NAA expression in DRG of CCI mice by metabolomics, suggesting that DRG neurons in NP mice may be damaged or dysfunctional. Studies have implied that NAA is synthesized from ASP in mitochondria, and then transported to the Schwann cells to synthesize myelin [23]. However, the decrease of NAA in DRG suggests a possible mechanism for demyelination in NP. NAAG is the third most common neurotransmitter in the nervous system behind glutamate and GABA [24]. It inhibits the release of glutamate through activation of metabotropic glutamate mGlu3 receptors at presynaptic membrane, and stimulates the release of neuroprotective growth factors through activation of mGlu3 receptors on glial cells [25]. Therefore, NAAG exerts neuroprotective effects by inhibiting excitotoxicity. And it has been confirmed that inhibition of NAAG degradation can treat the several animal models of disease, such as traumatic brain injury and neuropathic pain [25]. In this study, we found a significant decrease of NAAG in the DRG of the CCI mice, suggesting that it may reduce the inhibition of excitotoxicity to promote the progression of NP. Similarly, KEGG enrichment analysis showed significant changes in ala-nine, aspartate and glutamate metabolic pathways involved by NAA, NAAG and L-aspartate (Table 4).”

  1. Line 320: There should say “oleic acid…it suggests…”

Response:

Thanks a lot for your comments, we have revised the sentence to “Oleic acid, as a known endogenous neurotrophic factor, gets together with albumin and suggests neuroprotective effects by inhibiting nociceptive reflexes in SCI resulting in neuroprotective effects” on line 352-354, and marked as red.

  1. The authors need to explain the limitations of this study more in detail. Why are you using only males? Would you expect to find differences between males and females? would you consider exploring differences between male/men or female/women?

Response:

Thanks a lot for your comments, your advice is so helpful to us. We explain to you why we only use males. In clinical work, neuropathic pain is mainly measured by patient's subjective description and pain scale, but lacks objective indicators. Blood is the most commonly used test substance to react to the disease, which is less harmful to the human body. Therefore, one of the purposes of this study is to clarify whether there is a connection between DRG and serum metabolite changes in neuropathic pain mice, so as to find some clues of serum markers. While the effect of gender on pain and metabolites were not the focus of this study. Therefore, only male mice were used in the study. But the authors' advice is so good, and differences of metabolites between males and females with NP can be further studied in future experiments. According to your advice, we have described the limitations of this study on line 389--399, and marked as red. The content is as follows:

There are some limitations in our study, only male mice were used in this experiment to study the metabolite changes of the DRG and serum in neuropathic pain, and the effect of gender on pain and metabolites was not explored. And further experiments are needed to test this. In this study, we simply observed the results of metabolite changes in the DRG and serum of NP mice, but failed to further explore the reasons for metabolite changes caused by NP, which is a limitation of this experiment. It has been shown that dietary strategies influence pain behavior in patients with chronic pain [42, 43]. And lifestyle interventions may alter the gut microbiome to affect the pain response [44]. Therefore, we guess that in addition to the metabolic changes directly caused by nerve damage, the pain al-so affects the animal's behavior and diet, which in turn affects the metabolism. And this point needs to further study.

  1. Is the increased glucose measured in CCI animals caused by the sciatic nerve procedure? I understand that, in diabetic patients, the excess of glucose alters fatty acid biosynthesis, as has been shown, but the CCI procedure is not causing diabetes per se. Then, the question is what causes the increment in glucose. Can you extend your argument in your discussion?

Response:

Thanks a lot for your comments, your advice is so helpful to us. And we have further discussed the increased glucose of CCI animals on line 374-379, and marked as red. As you said, the CCI procedure is not causing diabetes per se, but the pain behavior may cause activity changes of the mice and then cause glucose metabolism disorders. It has been shown that brain glucose metabolism is increased in rats with chronic pain, and a clinical investigation showed a significant association between patients with chronic pain and diabetes. In our study, we observed a significant increase of serum glucose in CCI mice. Which may be caused by enhanced energy metabolism of the mice with pain behavior, or the change in eating habits caused by pain behavior. In addition, the increased serum glucose further aggravated the pain response in turn.

Reviewer 2 Report

Comments and Suggestions for Authors

The authors have produced an interesting and useful study to evaluate if the effects of : neuropathic pain on metabolites in serum and DRG were investigated using a non-targeted metabolomics approach detected by GC-MS and LC-MS to uncover differential metabolites and affected metabolic pathways associated with neuropathic pain.

However, I would like to make some observations.

1. Could the authors add the type of study in the title?

2. You must describe all acronyms before using them.

3. There is a systematic review on chronic widespread pain patients and lifestyle interventions that I think it would be interesting to suggest to the authors for discussion: DOI: 10.3390/medicina59020256

  • 4. Could the authors provide a graphical abstract of the study?
  •  
  •  
  • 5. There is a systematic review on chronic musculoskeletal pain and its conservative treatment that I think it would be interesting to suggest to the authors for discussion: DOI: 10.1097/PHM.0000000000002239
  •  

6. Could you add a section on "Clinical Implications"?

Comments on the Quality of English Language

No issues

Author Response

Dear Reviewer,

Thank you very much for your patient on our manuscript. Your advice is very helpful to us. According to your comments, the manuscript has been revised. Please find out our revised version of manuscript entitled with “Metabolomics analysis of DRG and serum in the CCI model of mice”. We have also made some other corrections and we hope that the present revised manuscript may make a better introduction to our present study.

  1. Could the authors add the type of study in the title?

Response:

Thanks a lot for your comments. According to your comments, we have supplemented the type of study on line 1, and marked as red.

  1. You must describe all acronyms before using them.

Response:

Thanks for your helpful comments. We have modified the incorrect use of abbreviations in the article and we have described the acronyms when we first use them on line 13, 17-20, 54-56, 60, 91, 100, 199, and marked as red.

  1. There is a systematic review on chronic widespread pain patients and lifestyle interventions that I think it would be interesting to suggest to the authors for discussion: DOI: 3390/medicina59020256

Response:

Thanks a lot for your comments. We have read this article carefully, it is a systematic review of the a correlation between lifestyle interventions and the gut microbiome in patients with chronic widespread pain (CWP). And it revealed that lifestyle interventions including exercise, electroacupuncture and a probiotic ingesting may improve the gut microbiota and reduce pain. In our study, we found a variety of differential metabolites in DRG and serum of NP mice. This review provides us a new perspective that metabolite changes in NP mice may be caused by dietary changes caused by pain behavior in addition to nerve injury, which may become a new direction of thinking. We have already discussed this point in the article on line 394-400, and marked as red.

  1. Could the authors provide a graphical abstract of the study?

Response:

Thanks a lot for your comments. We have added a graphical abstract of the study as fallows. And we have added it on page 3, and marked as red.

Figure 1. The flow chart of experiment.

  1. There is a systematic review on chronic musculoskeletal pain and its conservative treatment that I think it would be interesting to suggest to the authors for discussion: DOI: 10.1097/PHM.0000000000002239

Response:

Thanks a lot for your comments. We have read this article carefully.  This is a very interesting review with Meta-Analysis, and this review found that orthopedic manual therapy in isolation improved mechanical hyperalgesia although the quality of evidence is not perfect. This review gives us a new perspective that orthopedic manual therapy can also be considered for the treatment of chronic pain in addition to drugs, diet, electroacupuncture, etc.

  1. Could you add a section on "Clinical Implications"?

Response:

Thanks a lot for your comments. and we added the clinical implications on line 62-70, and marked as red. The content is as follows:

In clinical work, neuropathic pain is mainly measured by patient's subjective description and pain scale, but lacks objective indicators. Blood is the most commonly used test substance to react to the disease, which is less harmful to the human body. There have been numerous studies of DRG and serum organization in neuropathic pain, but few have explored metabolic changes in this condition and the connection between DRG and serum metabolites. Therefore, the purpose of this study is to explore the metabolite changes in DRG and serum and then clarify whether there is a connection between DRG and serum metabolite changes in neuropathic pain mice.  Which will provide us a new perspective for future clinical research.

Reviewer 3 Report

Comments and Suggestions for Authors

The authors conducted a study to characterize metabolic differences between mice undergoing nerve ligation (neuropathic model) vs sham. Please see my specific comment below but an overall summary of my requests is as follows.

(1) the justification for the study is poor and needs to be improved.

(2) the use of male mice only makes this study very weak and the authors need to provide a compelling justification for this.

(3) the altered metabolites may be due to the ligation (physiological responses to damage), the neuropathic mechanisms (the target in the paper), or the stress associated with the injury, etc. There is no discussion of any of these complexities or of any potential limitations of the study.

INTRODUCTION:

Simply stating that the underlying mechanisms of neuropathic pain are poorly understood does great disservice to the decades of research that has been done. We know a lot and this could be briefly explained.

Metabolomics is introduced as a technique (apparently in search of a problem) rather than introducing a problem (metabolites) that suggests a hypothesis in neuropathic pain that requires metabolomics approaches to answer it. Reading further, it really seems that the authors have a technique and they just want to see if it shows anything without any real justification or background hypothesis building.

Blood is not always the ultimate embodiment of metabolic changes as, for instance, in central pain the CSF may provide a better biofluid.

METHODS:

ARRIVE guidelines are not for animal welfare but they are a guide to reporting how this occurred. Please rethink this statement.

How old were the mice at the time of the experiment?

Why only males? The normal expectation is both sexes. This considerably weakens the study.

How was the hargreaves heat applied? Not described.

How was the DRG extracted? What levels? Bilateral or unilateral? How many DRGs typically went into 20mg? I assume that 7 days of surgery means 7 days of ligation. Why 7 days?

How did you get blood? How did you generate serum from blood? How was sample integrity ensured for all samples until freezing?

What were the determinants for significant difference (226 DRG and 350 serum) – p value and fold change? Please make sure that this is clearly explained throughout.

Fig 1 – are these t tests? But these are repeat measures are they not? Please explain what test is being done to determine the p values AND what the error bars are – SE or SD? Also please indicate if ALL animals were used or if you lost any or excluded any etc.

GENERAL COMMENTS

Use full names and abbreviations when first introducing – e.g. CCI.

English can be improved – e.g. some inapproriate use of “the” throughout. Generally, though, there are points that lack clarity and would benefit from revision by a native speaker/professional editor.

Comments on the Quality of English Language

Requires editing by a professional service or other native speaker.

Author Response

Dear Reviewer,

Thank you very much for your patient on our manuscript. Your advice is very helpful to us. According to your comments, the manuscript has been revised. Please find out our revised version of manuscript entitled with “Metabolomics analysis of DRG and serum in the CCI model of mice”. We have also made some other corrections and we hope that the present revised manuscript may make a better introduction to our present study.

  1. The justification for the study is poor and needs to be improved.

Response:

Thanks a lot for your comments. And we have improved the justification for the study on line 62-70, and marked as red. We hope to better explain to you the reasons for our research.

  1. The use of male mice only makes this study very weak and the authors need to provide a compelling justification for this.

Response:

Thanks a lot for your comments. This is a limitation of our experiment. we have described the limitations of this study on line 389-400, and marked as red. Next, we will explain why we use males. In clinical work, blood is the most commonly used test substance to react to the disease, which is less harmful to the human body. Therefore, one of the purposes of this study is to clarify whether there is a connection between DRG and serum metabolite changes in neuropathic pain mice, so as to find some clues of serum markers. While the effect of gender on pain and metabolites were not the focus of this study. Therefore, only male mice were used in the study. But the authors' advice is very good, and differences of metabolites between males and females with NP can be further studied in future experiments.

  1. The altered metabolites may be due to the ligation (physiological responses to damage), the neuropathic mechanisms (the target in the paper), or the stress associated with the injury, etc. There is no discussion of any of these complexities or of any potential limitations of the study.

Response:

Thanks a lot for your comments, your advice is so helpful to us. And we have added the discussion of the complexities and potential limitations of the study on line 389-400, and marked as red. The content is as follows:

There are some limitations in our study, only male mice were used in this experi-ment to study the metabolite changes of DRG and serum in neuropathic pain, and the effect of gender on pain and metabolites was not explored. And further experiments are needed to test this. In this study, we simply observed the results of metabolite changes in DRG and serum of NP mice, but failed to further explore the reasons for metabolite changes caused by NP, which is a limitation of this experiment. It has been shown that dietary strategies influence pain behavior in patients with chronic pain [44, 45]. And lifestyle interventions may alter the gut microbiome to affect the pain response [46]. Therefore, we guess that in addition to the metabolic changes directly caused by nerve damage, the stress associated with the nerve injury and the changes of animal's activity, energy metabolism and diet affected by pain behavior, which in turn affects the metabolites. And these points need to further study.

  1. How old were the mice at the time of the experiment?

Response:

The old of mice in our experiment is 8 weeks.

  1. How was the hargreaves heat applied? Not described.

Response:

Thanks a lot for your comments, and we have added the description of the hargreaves heat application on line 101-106, and marked as red.

  1. How was the DRG extracted? What levels? Bilateral or unilateral? How many DRGs typically went into 20mg? I assume that 7 days of surgery means 7 days of ligation. Why 7 days?

Response:

Thanks a lot for your comments, and we have added the details of the DRG collection on line 108-110, and marked as red. The reason why we chose 7 days of surgery is that the most pronounced pain response of CCI model is at about one week.

  1. How did you get blood? How did you generate serum from blood? How was sample integrity ensured for all samples until freezing?

Response:

Thanks a lot for your comments, and we have added the description of the serum collection on line 110-113, and marked as red.

The blood of each mouse was collected separately by removing the eyeball after anesthesia, and left at room temperature for 30 minutes, then the serum was the supernatant of the blood after 1200r centrifugation.

  1. What were the determinants for significant difference (226 DRG and 350 serum) – p value and fold change? Please make sure that this is clearly explained throughout.

Response:

Thanks a lot for your comments, p-value was obtained by t-test, and fold change is the ratio of the mean of the CCI group to the mean of the sham group. Significant differences are defined as P-values < 0.05 and FC > 1 or FC< 1.

  1. Fig 1 – are these t tests? But these are repeat measures are they not? Please explain what test is being done to determine the p values AND what the error bars are – SE or SD? Also please indicate if ALL animals were used or if you lost any or excluded any etc.

Response:

Thanks a lot for your comments. In figure 1, p-value was obtained by t-test and the error bars are SD, we only randomly selected the data of 6 mice from each group for statistics.

  1. Use full names and abbreviations when first introducing – e.g. CCI.

Response:

Thanks for your helpful comments. We have modified the incorrect use of abbreviations in the article and we have described the acronyms when we first use them on line 13, 17-20, 54-56, 60, 91, 100, 199, and marked as red.

  1. English can be improved – e.g. some inapproriate use of “the” throughout. Generally, though, there are points that lack clarity and would benefit from revision by a native speaker/professional editor.

Thanks for your helpful comments. We are so sorry for our poor English, and we selected the English editing company to modify the language and attached the modification certificate.

Reviewer 4 Report

Comments and Suggestions for Authors

Using gas chromatography-mass spectrometry (GC-MS) and liquid chromatography-mass spectrometry (LC-MS), the authors explored the metabolic alterations in dorsal root ganglia (DRG) and serum in a chronic constriction injury mouse model. Results of metabolomics analysis found that phospholipid, amino acid and acylcarnitine metabolic perturbations in DRG as well as phospholipid and glucose metabolic disorders in serum. This is an interesting study for metabolic impact of nerve injury.

 Some concerns this reviewer raised should be addressed by the authors.
1. The CCI model is used to induce neuropathic pain, and in the introduction and results, the authors mentioned the CCI-induced neuropathic pain. While the relationships between metabolic alterations and neuropathic pain were not discussed.

2. There is no normal control in this study. It is well known that surgery affects metabolism. The surgery procedures will result in severe injury in CCI groups compared to sham group.

3. Chloral hydrate has little or no analgesic effect and is not acceptable for anesthesia of small animals.

4. The conclusion that that NP caused dramatic changes in DRG and serum metabolites and the major metabolic pathway changes.  The pain will affect the animal behaviors and then metabolism, but there is lack evidence to support this conclusion. The effect of pain killer on metabolic pathway changes will be evidence for this conclusion.

5. GC-MS and LC-MS were used to detect the chemicals in serum and DRG. What is the difference of the results between these 2 techniques?   

6. Which and how many DRG were collected in each sample?

7. The quality of Figure 2 and Figure 3 should be improved.

8. The conclusion should be brief and clear.

Comments on the Quality of English Language

Spell out the full term at abbreviations first mention.

Author Response

Dear Reviewer,

Thank you very much for your patient on our manuscript. Your advice is very helpful to us. According to your comments, the manuscript has been revised. Please find out our revised version of manuscript entitled with “Metabolomics analysis of DRG and serum in the CCI model of mice”. We have also made some other corrections and we hope that the present revised manuscript may make a better introduction to our present study.

  1. The CCI model is used to induce neuropathic pain, and in the introduction and results, the authors mentioned the CCI-induced neuropathic pain. While the relationships between metabolic alterations and neuropathic pain were not discussed.

Response:

Thanks a lot for your comments. According to your comments, we have added the relationships between metabolic alterations and neuropathic pain on line 310-314, 328-333, 340-344, 356-361, and marked as red. Content is as follows:

The above results indicate that NP may affect mitochondrial function or myelin maintenance by inhibiting phospholipid synthesis in the DRG, studies have shown that structural abnormalities in either mitochondria or myelin sheaths can cause pain behavior [17, 18]. Therefore, NP may cause neuronal dysfunction and hyperalgesia by inhibiting phospholipid synthesis in DRG.

Additionally, some studies have shown that the NAA reduction was also observed in the brain of patients with neuropathic pain[20, 22], and the NAA level was positively correlated with the severity of neuropathic pain symptoms. In this study, we observed the decreased NAA expression in DRG of CCI mice by metabolomics, suggesting that DRG neurons in NP mice may be damaged or dysfunctional to cause a pain response.

NAAG exerts neuroprotective effects by inhibiting excitotoxicity. And it has been confirmed that inhibition of NAAG degradation can treat the several animal models of disease, such as traumatic brain injury and neuropathic pain [25]. In this study, we found a significant decrease of NAAG in the DRG of the CCI mice, suggesting that it may reduce the inhibition of excitotoxicity to promote the progression of NP.

Oleic acid inhibits the production of proinflammatory mediators in microglia cultures [35], and can attenuate pain response in patients with arthritis [36]. Therefore, oleic acid plays an important role in repairing nerve damage and attenuating pain responses. In this study, we found a significant increase in oleic acid in the CCI model, which could be a self-healing process after nerve injury, but its role in DRG needs further investigation.

  1. There is no normal control in this study. It is well known that surgery affects metabolism. The surgery procedures will result in severe injury in CCI groups compared to sham group.

Response:

Thanks a lot for your comments. The absence of a normal control group in this study is a limitation of our experiment, but the purposes of our experiment is to explore metabolic changes after nerve injury-induced pain behavior, while excluding the influence of other injury factors. Although the sham operation group has certain damage and metabolic changes, it can recover in a short time without obvious pain behavior. However, mice in the CCI group showed longer pain response and greater metabolic changes due to nerve injury. The metabolic differences of CCI in the sham operation group were compared to determine the metabolic changes caused by nerve injury-induced pain behavior.

  1. Chloral hydrate has little or no analgesic effect and is not acceptable for anesthesia of small animals.

Response:

Thanks a lot for your comments. And we will improve the anesthesia method in future animal experiments.

  1. The conclusion that that NP caused dramatic changes in DRG and serum metabolites and the major metabolic pathway changes. The pain will affect the animal behaviors and then metabolism, but there is lack evidence to support this conclusion. The effect of pain killer on metabolic pathway changes will be evidence for this conclusion.

Response:

Thanks a lot for your comments, and your comments were a great inspiration to us. This study simply observed the results of metabolite changes in DRG and serum of NP mice, but failed to further explore the reasons for metabolite changes caused by NP, which is a limitation of this experiment. As you said, in addition to the metabolic changes directly caused by nerve damage, the pain also affects the animal's behavior and diet, which in turn affects the metabolism. This may be one of the reasons for the metabolite changes caused by NP. And we discuss this point on line 389-400 and marked as red. These need us to explore furtherly in future experiments.

  1. GC-MS and LC-MS were used to detect the chemicals in serum and DRG. What is the difference of the results between these 2 techniques?

Response:

Thanks a lot for your comments. GC-MS is mainly used for direct qualitative and quantitative analysis of volatile components, and the separation and identification of the components can be completed at the same time. LC-MS is suitable for the analysis of non-volatile or poor thermal stability of metabolites. The combination of the two methods can obtain more comprehensive metabolites. The two methods use different internal reference as standard and measure the relative content of metabolites rather than absolute content, so it is necessary to compare GC-MS and LC-MS data separately between the two groups when performing multivariate and univariate statistical analysis.

  1. Which and how many DRG were collected in each sample?

Response:

Thank you so much for your comments. We have added the details of the sample collection on line 108-110, and marked as red. Content is as follows:

After 7 days of surgery, DRGs from the left L4-L6 segments of 10 mice in each group were collected together and used as one sample with a weight of approximately 20 mg.

  1. The quality of Figure 2 and Figure 3 should be improved.

Response:

Thank you so much for your comments. And we have revised the size of parameters and labels of Figure 3 (original Figure 2) and figure 4 (original Figure 3) on the page 6- 7, and marked as red.

  1. The conclusion should be brief and clear.

Response:

Thank you so much for your comments. And we have revised the conclusion on the line 404-414, and marked as red. The content is as follows:

Taken together, peripheral nerve injury results in significant alterations of phospholipid, amino acid, oleic acid and acylcarnitine in DRG, and alterations of glucose in serum. Furthermore, some metabolites showed opposite trends in DRG and serum, which implicated a close relationship between DRG and serum metabolites, and serum metabolites could reflect the metabolic changes in DRG. In addition, the reasons of metabolite changes could be as follows: direct nerve damage, the stress associated with the nerve injury and the changes of animal's activity, energy metabolism and diet affected by pain behavior. Moreover, the changed metabolites can reflect cellular structural disruption or abnormal energy metabolism, as well as postinjury self-protection. There are some limitations to this experiment, and structural and functional changes in tissues still need to be validated by more histomorphology and molecular biology studies.

Round 2

Reviewer 2 Report

Comments and Suggestions for Authors

The authors have improved the previous version of their manuscript with the current version.

Congratulations

Author Response

Dear reviewer,

Thank you very much for your recognition of our research, and your valuable comments greatly helped us to improve the article.

Very sincerely yours,

Lina Huang

Institution and address: Department of Anesthesiology, Shanghai General Hospital, Shanghai Jiao Tong University of Medicine, Shanghai, 20080, China;

Tel: 18001780050

Reviewer 4 Report

Comments and Suggestions for Authors

The author found metabolic changes in DRG and serum form CCI mouse, which is induced buy nerve injury. The neuropathic pain may result in and/or result from metabolic changes, however there is no evidence to confirm the conclusion "study found that NP caused dramatic changes in the DRG and serum metabolites and the major meta- 399 bolic pathway changes." (line 398-399).

Comments on the Quality of English Language

 Minor editing of English language 
